# Regionalization for health improvement: A systematic review

**Maíra Catharina Ramos** [1,2] *, **Jorge Otávio Maia Barreto**[2], **Helena Eri Shimizu**[1‡],
**Amanda Pereira Gomes de Moraes**[3‡], **Everton Nunes da Silva**[4]

**1** Faculty of Health Sciences, University of Brasilia, Brasília, Brazil, **2** Oswaldo Cruz Foundation, Brasília, Brazil, **3** Department of Medicine, Federal University of Sergipe, Lagarto, Brazil, **4** Faculty of Ceilândia, University of Brasilia, Brasília, Brazil

☯ These authors contributed equally to this work.
‡ HES and APGM also contributed equally to this work.
* maira.ramos@fiocruz.br

## Abstract

Regionalization is the integrated organization of a healthcare system, wherein regional structures are responsible for providing and administrating health services in a specific region. This method was adopted by several countries to improve the quality of provided care and to properly utilize available resources. Thus, a systematic review was conducted to verify effective interventions to improve health and management indicators within the health services regionalization. The protocol was registered in PROSPERO (CRD42016042314). We performed a systematic search in databases during February and March 2017 which was updated in October 2020. There was no language or date restriction. We included experimental and observational studies with interventions focused on regionalization-related actions, measures or policies aimed at decentralizing and organizing health offerings, rationalizing scarce capital and human resources, coordinating health services. A methodological assessment of the studies was performed using instruments from the Joanna Briggs Institute and GRADE was also used to assess outcomes. Thirty-nine articles fulfilled the eligibility criteria and sixteen interventions were identified that indicated different degrees of recommendations for improving the management of health system regionalization. The results showed that regionalization was effective under administrative decentralization and for rationalization of resources. The most investigated intervention was the strategy of concentrating procedures in high-volume hospitals, which showed positive outcomes, especially with the reduction of hospitalization days and in-hospital mortality rates. When implementing regionalization, it must be noted that it involves changes in current standards of health practice and in the distribution of health resources, especially for specialized services.

## Introduction

Health regionalization has been adopted by several countries to improve population access to healthcare services [1, 2]. The World Health Organization (WHO) defines it as the rational

---

**Competing interests:** The authors have declared no conflict of interest.

distribution of medical services throughout the territory, ensuring that services and facilities provide all levels of care (primary, secondary and tertiary) at easy access to the population and cost-effective care [1].

From this perspective, regionalization contributes to the process of organizing the health service network at regional or local level by decentralizing certain geographical areas as well as the concentration of health activities under a single command to achieve better health conditions, providing full accessibility and full coverage to the population. In addition, this process makes it possible to decide which political and administrative division to implement by establishing new internal borders in order to facilitate the exercise of power and to adapt to a new decentralized public administration [2].

The literature shows the Dawson Report as one of the first and main documents to concern regionalization in 1920. The author pictured an integrated healthcare system wherein the prevention of diseases and healthcare promotion would be coordinated, including primary, secondary and tertiary care of both short and long term serving to organize health services in territories specifically selected for such services [3]. England, Canada and Brazil are amidst the countries that used Dawson's ideals to develop health services regionalization policies to integrate the coordination of institutional and community integration in order for the community to participate, and to transfer the authority of management concerning healthcare programs to regional level. The centralization of governance by eliminating local healthcare entities and agencies in these countries was followed by the decentralization of responsibility towards providing most health services [4]. Likewise, there is no consensus on the concept of regionalization. In federative countries, the concept that is closest to regionalization is: the integrated organization of a healthcare system with several coordinated functions that serves a specific territory wherein a regional structure would be responsible to provide and administrate healthcare services [5, 6].

According to Marchildon [7] the goal of regionalization is to integrate and rationalize healthcare services, to promote evidence-based practice, to decentralize resources and decision-making, and to switch focus and resources towards the prevention of illnesses and promotion of health. For regionalization to take place, a number of basic requirements are necessary: to regulate the responsibility of each level and its units, to match supply with the demand of the population, to establish the gateway to the regionalized system, to establish a referral system of the lowest to the highest level, to establish a continuous flow of cross-information, to establish scientific and technical support mechanisms, and to establish a partnership between health professionals and technicians [8].

Studies show that regionalization needs organizational arrangements to achieve its proper functioning [9, 10], namely: i) coordination: concerns the integration, sharing, articulation and decision-making based on norms, legal or not, whose actors agree to share decisions and tasks; in forums and political mechanisms for intergovernmental negotiation; the functioning of representative institutions; in the coordinating and/or inducing role of Brazilian government [11]; ii) decentralization: defined as the delegation or devolution of certain responsibilities and functions, broadening the view on the autonomy of other federative entities [12]; iii) rationalization: it is defined as a resource utilization tool aiming to increase the efficiency of services without bias towards the effectiveness of the actions [13]; iv) governance: it is defined as the actions and means adopted by society to organize itself to protect and promote the health of the population, seeking to harmonize the decisions and actions of different actors in favor of the equity and sustainability of health systems [14].

Some countries have been successful in fully or partially adopting these dimensions in the regionalization model [11, 12], but studies showing which interventions are effective in improving health as well as management indicators in this process are still rare.

This study thus sought to identify and characterize effective interventions to improve health and management indicators within the scope of the regionalization of health services. A systematic review was conducted to answer the following question: what are effective interventions to improve health and management indicators within health services regionalization?

## Methodology

A systematic review was conducted to answer the following question: what are effective interventions to improve health and management indicators concerning health services regionalization? From the structured question, the PICOT question was defined, where: P) health services supply arrangements; I) Interventions aimed at regionalization in terms of actions, measures or policies aimed at decentralizing and organizing health offerings, rationalizing scarce capital and human resources, coordinating health services; C) Interventions that do not take into account any dimension of regionalization; O) Primary: any health outcome (mortality rate, infant mortality rate, preventable mortality rate, ambulatory care-sensitive conditions, quality of life, comorbidity) and secondary: any effect on management (governance arrangements; financial arrangements; service delivery arrangements; implementation interventions); T) experimental studies (randomized, *quasi*-randomized and clinical trials) and observational studies (longitudinal, cohort, case-control and cross-sectional studies). The Taxonomy of Decisions of Universidade McMaster was used to define effects on management. This taxonomy allows for better homogenization of health descriptors, as it proposes useful patterns for ideal decision-making and provides tools to take good practical decisions [15].

We searched the MEDLINE databases via Pubmed, Virtual Health Library (VHL), The Cochrane Library (Cochrane Database of Systematic Reviews, Cochrane Central Register of Controlled Trials (CENTRAL), Cochrane Methodology Register), PDQ, Science Direct, Scopus, Web of Science, Center for Reviews and Dissemination (CRD), Google Scholar, Health evidence and Health Systems Evidence via EVIPNet.

The following search strategy was used: ("regional health planning" OR "regional governments" OR "regional healthcare" OR "area wide planning" OR "regional healthcare network" OR "comprehensive health planning" OR "annual implementation plans" OR "regional governments" OR "regionalization") AND ("decentralization" OR "coordination" OR "co ordination" OR "rationalization" OR "rationaliz$" OR "governance).

There was no language or date restriction, and the search was conducted between February and March 2017 and updated in October 2020.

The inclusion criteria were studies that answered the PICOT question. Studies concerning systematic reviews methodology, overviews of systematic reviews, theoretical, qualitative, editorial studies and letters to the editor were excluded, as well as studies that did not address interventions related to clearly-defined organizational arrangements of regionalization, or that did not provide the full text of the study.

The systematic review followed the PRISMA for systematic review protocols. The protocol was previously registered on the Prospero platform (CRD42016042314).

The selection of studies was performed in two steps: i) selection by title and abstract by four independent reviewers (MR, ENS, JOMB and HES), and the differences resolved by consensus; ii) selection after reading of the studies in full by two independent reviewers (MR and ENS), with disagreements resolved by consensus.

For data extraction, a table was used to collect the following information: author and year, publication country of origin, research method, sample, intervention, main results and area of publication. All extraction was performed by two independent evaluators (MR and ENS).

A methodological assessment of the studies was performed using instruments from the Joanna Briggs Institute of the University of Adelaide, Australia (available at https://joannabriggs.org/critical_appraisal_tools). The Grading of Recommendations Assessment, Development and Evaluation (GRADE) was adopted to evaluate the results. The GRADE system is a transparent structure for developing and presenting evidence summaries, providing a systematic approach for recommendations to improve clinical practice, with clear and concise information over the quality of evidence (the degree to which a given result might be trusted) and the reliability of the recommendation [16]. With GRADE, the interventions identified during systematic review were stratified as "Level of recommendation: very low recommendation; low recommendation; moderate recommendation, and; high recommendation".

## Results

The search found 3921 documents, of which 604 were duplicates. After reading the titles and abstracts, 601 documents were excluded. The reason for excluding this step is described in S1 Appendix. After complete reading of the documents, 39 studies were selected to compose the systematic review. Fig 1 describes this process.

Table 1 summarizes the main characteristics of the studies included in this systematic review. Regarding the year of publication, there is a study published in the 1980s [42], six in the 1990s [30, 31, 33–35, 38], seventeen in the 2000s [17–20, 27–29, 32, 37, 41, 43, 46, 47, 52, 54–56], fourteen in the 2010s [22–24, 26, 36, 39, 40, 44, 45, 48–51, 53] and one in the 2020s [21]. As for the period studied, there is a wide variety. Studies [17, 29, 32, 37, 41, 42, 46, 50, 51] with up to one year of analysis corresponded to 23.07% (n = 9) of publications, followed by those with more than 10 years [20, 24, 31, 55] (10.25%; n = 4) and those with one year of analysis [27, 33, 38, 56] (10.25%; n = 4). Regarding the country of origin, 56.41% (n = 22) are from the United States [18, 20, 22, 23, 27, 29–31, 35–39, 41–43, 45–47, 52, 55, 56]. Publications from Italy [21, 26, 48] total 7.69% (n = 3), while publications from Australia [32, 51], Brazil [17, 40], the Netherlands [50, 54] and Canada [28, 34] account for 5.12% (n = 2) each. Zambia [19], Uganda [49] and a partnership between Canada and the United States [33], and between Italy and Spain [24], each of the publications with 2.56%. (n = 1).

Regarding the study method, 11 distinct methods were identified, and the most frequent was the regression model [17, 20–22, 24, 27, 29–31, 33, 35–38, 42, 43, 46, 47, 52–56], in general, with 58.97% (n = 23). However, it is noteworthy that the regression models varied between simultaneous, Poisson, linear, generalized linear equations with Poisson error structure, logistical, logistical and linear, hierarchical, multiple and Cox multivariate logistics. Ecological [32, 37, 48] and cross-sectional studies [40, 49, 51] add up to 7.69% each (n = 3) and quasi-experimental studies [44, 45] accounted for 5.12% (n = 2). The methods of cost-utility analysis [18], proportional risk survival analysis [28], cost-effectiveness analysis [23], cost analysis [50], Propensity score [41], multiple comparison test [34], cohort [39] and Data Envelopment Analysis [26] totaled 2.56% of the studies, each.

For quality assessment, the instrument of The Joanna Briggs Institute was used [57]. After weighting by two independent researchers, 58.97% of the studies obtained maximum quality assessment (8/8), while 23.07% obtained intermediate evaluation (7/8 or 6/8). Another 10.25% of the studies had regular evaluation (4/8) and 5.12% poor evaluation, with 0/8 points established by the instrument. The result of the quality assessment can be seen in detail in Table 2.

In addition to evaluating the methodological quality of the included studies, we chose to assess the degree of recommendation of evidence by outcome. To this end, the Grade System (Grading of Recommendations Assessment, Development and Evaluation) [58] of assessment (Table 3) was used. Interventions of the coordination dimension were assessed

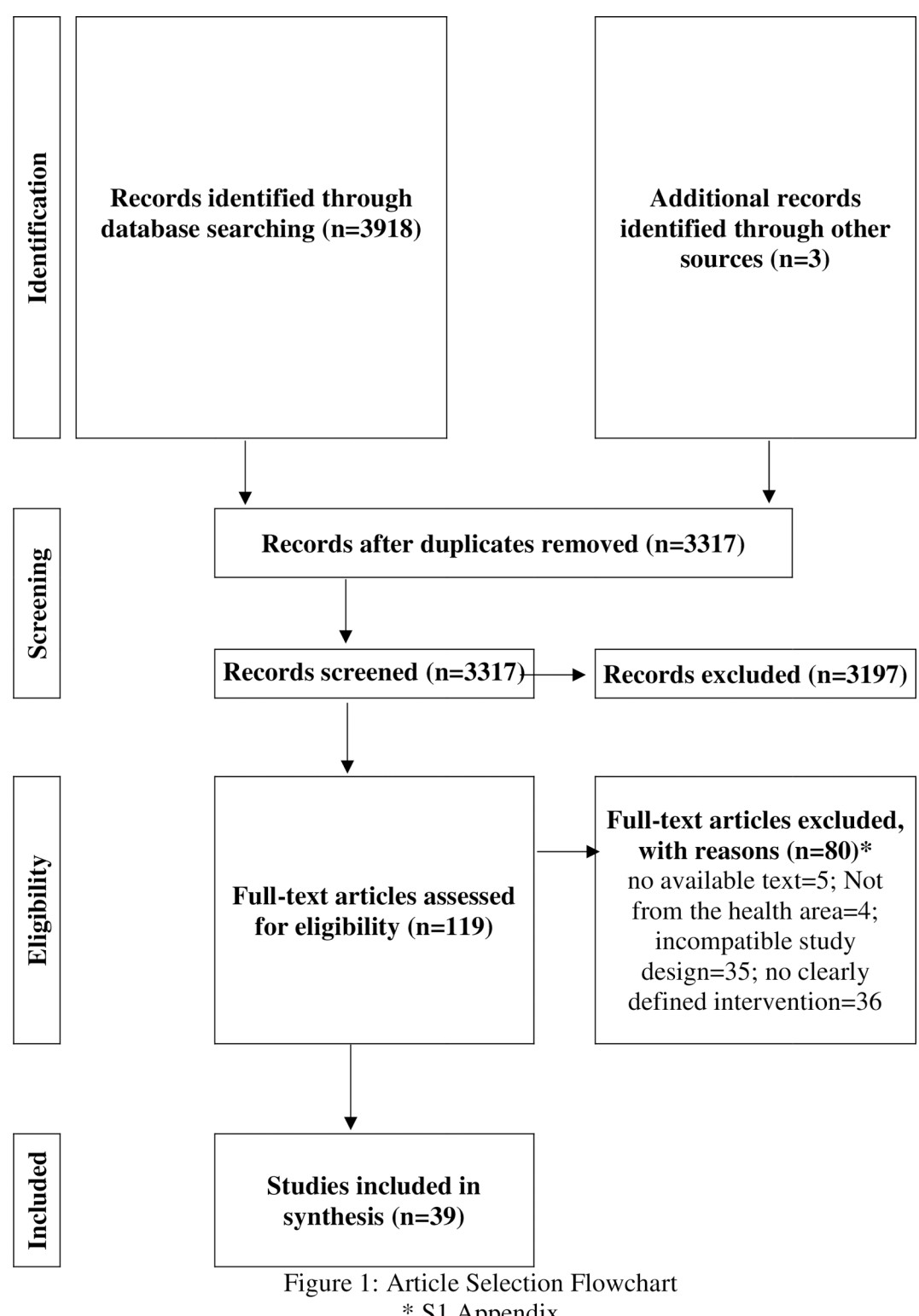

Figure 1: Article Selection Flowchart
* S1 Appendix

**Fig 1. Article selection flowchart.** * S1 Appendix.

**Table 1. Included studies by identified dimension.**

| Study | Country | Method | Period of analyzed data | Dimension |
|---|---|---|---|---|
| Atkinson & Haran (2004) [17] | Brazil | Multiple regression model | Oct-Dec/1997 | Decentralization |
| Bardach et al. (2004) [18] | USA | Cost-utility study | 1990–1998 | Rationalization |
| Bossert, Chitah & Bowser (2003) [19] | Zambia | Ecological study | 1995–1998 | Decentralization |
| Brookfield et al. (2009) [20] | USA | Cox multivariate regression model | 1990–2000 | Rationalization |
| Cavalieri & Ferrante (2020) [21] | Italy | Linear regression model | 1996–2016 | Decentralization |
| Chen et al. 2018 [22] | USA | Multivariate regression model | 2004–2013 | Decentralization |
| Concannon et al. (2014) [23] | USA | Cost-effectiveness | 1996–1999 | Rationalization |
| Costa-front 2018 [24] | Spain-Italy | Linear regression model | 1998–2009 | Decentralization |
| Cowan et al. (2003) [25] | USA | Logistic Regression Model | 1996–1997 | Rationalization |
| De Nicola et al. (2005) [26] | Italy | Data Envelopment Analysis and Truncated Regression | 2004–2005 | Decentralization |
| Dimick et al. (2002) [27] | USA | Logistic and linear regression model | 1996–1997 | Rationalization |
| Dueck et al. (2004) [28] | Canada | Proportional risk survival analysis | 1992–2001 | Rationalization |
| Glance et al. (2002) [29] | USA | Logistic Regression Model | 1999 | Rationalization |
| Glasgow et al. (1999) [30] | USA | Multiple regression model | 1990–1994 | Rationalization |
| Gordon et al. (1998) [31] | USA | Poisson regression model | 1984–1995 | Rationalization |
| Gruen, Weeramanthri & Bailie (2002) [32] | Australia | Ecological | 2000 | Coordination |
| Grumbach et al. (1995) [33] | USA, Canada | Linear regression model | 1987–1989 | Rationalization |
| Hamilton et al. (1997) [34] | Canada | Multiple comparison test | 1992–1996 | Rationalization |
| Imperato et al. (1997) [35] | USA | Logistic Regression Model | 1991–1994 | Rationalization |
| Jollis 2018 [36] | USA | Linear regression model | 2015–2017 | Decentralization |
| Ko et al. (2002) [37] | USA | Multivariate Logistic Regression Model | 1996 | Rationalization |
| Konvolinka, Copes & Sacco (1995) [38] | USA | Logistic Regression Model | 1988–1989 | Rationalization |
| Lau et al. (2014) [39] | USA | Cohort | 2005–2013 | Rationalization |
| Lima (2010) [40] | Brazil | Cross-sectional | 2000–2006 | Coordination |
| MacKenzie et al. (2006) [41] | USA | Propensity score | 1999 | Rationalization |
| Maerki, Luft & Hunt (1986) [42] | USA | Regression model of simultaneous equations | 1972 | Rationalization |
| Marcin et al. (2008) [43] | USA | Hierarchical Logistic Regression Model | 1998–2002 | Rationalization |
| Maritaz et al. (2019) [44] | France | Quasi-experimental | 2013–2016 | Coordination |
| Muoto et al. (2016) [45] | USA | Quasi-experimental | 2008–2012 | Coordination |
| Nallamothu et al. (2001) [46] | USA | Logistic Regression Model | 1997 | Rationalization |
| Nathens et al. (2001) [47] | USA | Logistic and linear regression model | Nov/1997-Jul/1998 | Rationalization |
| Nuti et al. (2016) [48] | Italy | Ecological study | 2007–2012 | Governance |
| Okello et al. (2018) [49] | Uganda | Cross-sectional | 2010–2016 | Decentralization |
| Tanke & Ikkersheim (2012) [50] | Netherlands | Cost analysis | 07/2011 | Rationalization |
| Turner, Mulholland & Taylor (2011) [51] | Australia | Cross-sectional study | Jan-April/2009 | Coordination |
| Urbach, Bell & Austin (2003) [52] | USA | Logistic Regression Model | 1994–1999 | Rationalization |
| Valdes-Stauber et al. (2014) [53] | Germany | Multivariate regression model | 2002–2010 | Decentralization |
| Vernooij et al. (2008) [54] | Netherlands | Generalized linear regression model with Poisson error structure | 1996–2003 | Rationalization |
| Waines et al. (2003) [55] | USA | Linear regression model | 1988–2000 | Rationalization |

as "moderate recommendation" and "very low recommendation" as well as interventions and "Public Disclosure" (governance). The decentralization dimension showed an evaluation of "high" and "moderate recommendation", in addition to the "very low recommendation". The dimension of rationalization showed an evaluation of "moderate" and "high recommendation" (Table 3).

**Table 2. Methodological quality assessment of included studies.**

| Study | Were the sample inclusion criteria clearly defined? | Were the research subjects and study characteristics described in detail? | Was exposure validly and reliably measured? | Were the objective and standard criteria used to measure the condition? | Have confounding factors been identified? | Have strategies been addressed to deal with stated confounders? | Were results validly and reliably measured? | Was statistical analysis used appropriately? | Total |
|---|---|---|---|---|---|---|---|---|---|
| Atkinson & Haran (2004) [17] | Yes | Yes | Yes | Yes | Yes | Yes | Yes | Yes | 8/8 |
| Bardach et al. (2004) [18] | Yes | Yes | Yes | Yes | Yes | Yes | Yes | Yes | 8/8 |
| Brookfield et al. (2009) [20] | Yes | Yes | Yes | Yes | Yes | Yes | Yes | Yes | 8/8 |
| Chen et al. (2018) [22] | Yes | Yes | Yes | Yes | Yes | Yes | Yes | Yes | 8/8 |
| Concannon et al. (2014) [23] | Yes | Yes | Yes | Yes | Yes | Yes | Yes | Yes | 8/8 |
| Cowan et al. (2003) [25] | Yes | Yes | Yes | Yes | Yes | Yes | Yes | Yes | 8/8 |
| De Nicola et al. (2005) [26] | Yes | Yes | Yes | Yes | Yes | Yes | Yes | Yes | 8/8 |
| Dimick et al. (2002) [27] | Yes | Yes | Yes | Yes | Yes | Yes | Yes | Yes | 8/8 |
| Dueck et al. (2004) [28] | Yes | Yes | Yes | Yes | Yes | Yes | Yes | Yes | 8/8 |
| Glance et al. (2002) [29] | Yes | Yes | Yes | Yes | Yes | Yes | Yes | Yes | 8/8 |
| Glasgow et al. (1999) [30] | Yes | Yes | Yes | Yes | Yes | Yes | Yes | Yes | 8/8 |
| Gordon et al. (1998) [31] | Yes | Yes | Yes | Yes | Yes | Yes | Yes | Yes | 8/8 |
| Jollis et al. (2018) [36] | Yes | Yes | Yes | Yes | Yes | Yes | Yes | Yes | 8/8 |
| Ko et al. (2002) [37] | Yes | Yes | Yes | Yes | Yes | Yes | Yes | Yes | 8/8 |
| Konvolinka, Copes & Sacco (1995) [38] | Yes | Yes | Yes | Yes | Yes | Yes | Yes | Yes | 8/8 |
| MacKenzie et al. (2006) [41] | Yes | Yes | Yes | Yes | Yes | Yes | Yes | Yes | 8/8 |
| Maerki, Luft & Hunt (1986) [42] | Yes | Yes | Yes | Yes | Yes | Yes | Yes | Yes | 8/8 |
| Marcin et al. (2008) [43] | Yes | Yes | Yes | Yes | Yes | Yes | Yes | Yes | 8/8 |
| Muoto et al. (2016) [45] | Yes | Yes | Yes | Yes | Yes | Yes | Yes | Yes | 8/8 |
| Nallamothu et al. (2001) [46] | Yes | Yes | Yes | Yes | Yes | Yes | Yes | Yes | 8/8 |
| Nathens et al. (2001) [47] | Yes | Yes | Yes | Yes | Yes | Yes | Yes | Yes | 8/8 |
| Urbach, Bell & Austin (2003) [52] | Yes | Yes | Yes | Yes | Yes | Yes | Yes | Yes | 8/8 |
| Vernooij et al. (2008) [54] | Yes | Yes | Yes | Yes | Yes | Yes | Yes | Yes | 8/8 |

*(Continued)*

**Table 2.** (Continued)

| Study | Were the sample inclusion criteria clearly defined? | Were the research subjects and study characteristics described in detail? | Was exposure validly and reliably measured? | Were the objective and standard criteria used to measure the condition? | Have confounding factors been identified? | Have strategies been addressed to deal with stated confounders? | Were results validly and reliably measured? | Was statistical analysis used appropriately? | Total |
|---|---|---|---|---|---|---|---|---|---|
| Wainess et al. (2003) [55] | Yes | Yes | Yes | Yes | Yes | Yes | Yes | Yes | 8/8 |
| Cavalieri & Ferrante (2020) [21] | Yes | Yes | Yes | Yes | Yes | No | Yes | Yes | 7/8 |
| Imperato et al. (1997) [35] | No | Yes | Yes | Yes | Yes | Yes | Yes | Yes | 7/8 |
| Costa-Font et al. (2018) [24] | Yes | Yes | Yes | Yes | No | No | Yes | Yes | 6/8 |
| Grumbach et al. (1995) [33] | Yes | Yes | Yes | Yes | Yes | Yes | No | No | 6/8 |
| Lau et al. (2014) [39] | Yes | Yes | Yes | Yes | Unclear | Unclear | Yes | Yes | 6/8 |
| Maritaz et al. (2019) [44] | Yes | Yes | Yes | Yes | No | No | Yes | Yes | 6/8 |
| Okello et al. (2018) [49] | Yes | Yes | Yes | Yes | No | No | Yes | Yes | 6/8 |
| Tanke & Ikkersheim (2012) [50] | Yes | Yes | Unclear | Yes | Unclear | Yes | Yes | Yes | 6/8 |
| Valdes-Stauber et al. (2014) [53] | Yes | Yes | Yes | Yes | No | No | Yes | Yes | 6/8 |
| Bossert, Chitah & Bowser (2003) [19] | Yes | Yes | Yes | Yes | No | No | No | No | 4/8 |
| Lima (2010) [40] | Yes | Yes | Yes | Yes | No | No | No | No | 4/8 |
| Hamilton et al. (1997) [34] | No | No | Yes | Yes | No | No | Yes | Yes | 4/8 |
| Nuti et al. (2016) [48] | Yes | Yes | Yes | Yes | No | No | No | No | 4/8 |
| Gruen, Weeramanthri & Bailie (2002) [32] | No | No | No | No | No | No | No | No | 0/8 |
| Turner, Mulholland & Taylor (2011) [51] | No | No | Unclear | No | No | No | No | No | 0/8 |

In all, 16 interventions were identified and divided into 4 dimensions: coordination, decentralization, rationalization and governance.

Within the "coordination" dimension were the interventions of "Integration of surgery and cancer therapy in the same physical facility", "Provision of sanitary transport services between areas further from the high-volume hospital", "Coordinated care organization", "Outreach services in remote areas" and "Outpatient and psychosocial network integration." The first intervention is about integrating cancer surgery with chemotherapy in the same physical setting in cases of head and neck cancer [22]. Patients with fragmented care were found to be associated with poorer overall survival, regardless of the other variables assessed in the study (RR 1.08; 95% CI 1.03–1.13).

**Table 3. Evaluation of regionalized interventions located in the systematic review, by dimension.**

| Dimension | Intervention | Expected effect | Outcome | Number of articles corroborating the effect | Number of articles that do not corroborate the effect | Number of articles with mixed evidence | Level of recommendation* |
|---|---|---|---|---|---|---|---|
| A. Coordination | A1. Integration of surgery and cancer therapy in the same physical facility | + | Increased overall survival | 1 | - | - | moderate recommendation |
| | A2. Provision of health transportation services between areas further from the polo hospital | + | Reduction of waiting time for attendance | 1 | - | - | moderate recommendation |
| | A3. Coordinated care organization | + | Increased health care and improved quality of life | 1 | - | 1 | low recommendation |
| | A4. Outreach service in remote areas | + | Increased surgical production and clinical performance. | 2 | - | - | very low recommendation |
| | | | Reduction of waiting time and attendance costs | | | | very low recommendation |
| | A5. Outpatient and psychosocial network integration | + | Reduction in number of days of hospitalization | 1 | - | - | very low recommendation |
| B. Decentralization | B1. Decentralization of care | + | Increased health care | 3 | - | 1 | high recommendation |
| | B2. Allow patient mobility between regions | + | Increased efficiency of a healthcare organization | 1 | - | - | moderate recommendation |
| | B3. Freedom of user choice between public and private providers | + | More efficient organizational model in health | 1 | - | - | moderate recommendation |
| | B4. National Reimbursement System considering regional characteristics of population and structure of health system | + | Increased health system efficiency | 1 | - | - | moderate recommendation |
| | B5. Fiscal decentralization | ± | Decrease in infant mortality rate and increase in life expectancy at birth | | | 1 | low recommendation |
| | B6. Vertical centralized programs in decentralized context | - | Decreased program efficiency | 1 | - | - | very low recommendation |
| C. Rationalization | C1. Provision of health transportation services between areas further from a high-volume hospital | + | They are more cost-effective than building new health facilities | 1 | - | - | moderate recommendation |
| | C2. Concentration of procedures in hospitals with high volume of production | + | Reduction in number of days of hospitalization | 19* | 1 | 1 | high recommendation |
| | | | Reduction of hospital mortality rate | | | | high recommendation |
| | C3. Transfer of patients arriving from low-volume hospital to high-volume hospitals | + | Increased survival rate | 1 | - | - | moderate recommendation |
| | C4. Concentration of procedures in specialized hospitals | + | Increased survival rate | 1 | - | - | moderate recommendation |

*(Continued)*

**Table 3.** (Continued)

| Dimension | Intervention | Expected effect | Outcome | Number of articles corroborating the effect | Number of articles that do not corroborate the effect | Number of articles with mixed evidence | Level of recommendation* |
|---|---|---|---|---|---|---|---|
| D. Governance | D1. Public disclosure of data | + | Improved health system outcomes | 1 | - | - | very low recommendation |

Notes:

B2, B3, B4 refer to the same study

* One study showed positive results regarding intervention, but with reservations

Legend:

+ positive effect

- negative effect

The second intervention of this dimension was the transport of patients to referral hospitals [36]. The waiting time for care was significantly reduced, where at least 75% of patients had coronary artery disease treatment within 90 minutes (p<0.0001). Hospital mortality fell from 4.4% to 2.3% (p = 0.001) and heart failure as a complication fell from 7.4% to 5.0% (p = 0.031). After adjusting for demographic and clinical characteristics, mortality remained statistically significant, with an odds ratio of 2.16 (95% CI, 1.17–3.99; p = 0.013).

The third intervention refers to coordinated care organization, which are patient-centered integrated health services that are geographically defined with a reference site for population care. They are services of low and medium complexity that act as a gateway to outpatient care. Two different studies have been identified for depicting this type of intervention [44, 45].

The first one, carried out in France, performed a quasi-experimental study devised into two clusters: 'everyday care' and 'coordinated care' in the treatment of cancer patients who were on oral medication. The authors identified that the overall six-months survival rate was of 76% in the first cluster and of 87% in the latter (p = 0.064). No significant statistical changes were observed in the rates of disease progression, quality of life or treatment compliance [44].

The second study was conducted in the United States of America (USA). It analyzed the trends in early prenatal care initiation and trends in prenatal care adequacy after the implementation of the coordinated care organization model. The authors found that the rate of early prenatal care initiation increased significantly, from 73.1% to 77.3%, while prenatal care adequacy from 65.9% to 70.5%. The effect of implementing CCO in prenatal care adequacy, however, was not significant. After a sensitivity analysis, the estimates remained constant [45].

The fourth intervention refers to the possibility of a specialist doctor or group of specialist doctors going to communities in remote and vulnerable areas to make regular visits, and may even use telemedicine in more urgent cases, to help the referral team. The two studies identified in this intervention come from Australia, which suffers from a shortage of doctors in remote areas. Both showed a positive effect on health, increasing surgical production by 1.9 times and clinical yields by 1.4 times. Waiting time for service was also reduced by 42% when there were 5 or more coordination measures with the outreach service [51]. Cost savings were U$277 compared to conventional services, which cost AU$450 [32]. Although studies indicate a positive effect, both received poor quality assessment, being 0/7 in the study by Turner, Mulholland & Taylor [51] and 0/8 by Gruen, Weeramanthri & Bailie [32]. In assessing grade recommendation, intervention outcomes received "very low degree of recommendation" due to methodological limitations and did not present any factor that increased outcome recommendation.

The fifth intervention of the coordination dimension was called "Outpatient and psychosocial network integration". According to the author [40], the operation of outpatient and psychosocial services in the network reduces the hospitalization time of the patient in crisis from 24 to 9 days, however, no confounding factors that could be influencing the results were controlled, and statistical analysis was not performed, and for these reasons the resulting evaluation was 4/8 in terms of study quality. After evaluating the outcome of the intervention, its degree of recommendation was very low, since in addition to methodological limitations, the outcome is inconsistent and has no factors that could increase its recommendation, as shown in Table 3.

Six interventions were categorized as belonging to the "decentralization" dimension. The first, "Decentralization of care", states that health care actions should be performed at the regional level, including four studies in this intervention. In a study with good methodological quality conducted by Atkinson & Haran [17] comparing prenatal care in Ceará in completely decentralized hospitals to centralized hospitals, a 23.8% increase in prenatal care was present. In another identified study [49], the authors found that the decentralization of care for rheumatic diseases in Uganda decreased the severity of the disease in the population served (p 0.20). Prior to the regionalization process, the proportion of patients with severe rheumatic diseases was 74.4%, decreasing to 72.7% after regionalization was implemented, while mild cases increased from 9 to 12.4%.

The third study included assessed user satisfaction and public expenditure per capita in Spain and Italy between 1998–2002 and 2003–2009, the last period after the regionalization process [24]. In the case of Italy, inequality in spending actually declined in the second period (p 0.01), but process-related quality inequality, measured against user satisfaction, did not vary significantly; thus, decentralization produced differences in spending but not in outcomes (satisfaction). After decentralization, both countries decreased inequality in fiscal capacities (p 0.01), with such decrease being higher in Spain. After the Oaxaca-Blinder decomposition method, the authors state that the consolidation of federalism seems to influence the way spending is transformed into procedural outcome; however, the region's history of autonomy is relevant. If, prior to regionalization, the municipality already had high levels of autonomy, then there is no statistical difference between regionalizing or not, according to the authors.

The last study included in this intervention compared the health outcomes of patients who were tended by either traditional or decentralized teams. The analyzed supply variables showed that: patients who had been tended by decentralized teams experienced only half of the expected amount of treatment and of the annual outpatient doctor-patient interactions; but the amount of hospitalizations was 17.3% higher among patients who had been given decentralized care. The annual costs of hospitalizations ranged from 886 to 3223 Euros for decentralized teams and 2038 Euros for central outpatient care. The costs of psychotropic and outpatient treatment, however, are significantly higher for the central outpatient department (1,050 Euros per year for the former and 781 Euros for the latter). The authors conclude that the number of persons admitted has slightly increased, as did the number of admissions. Despite that, the number of admissions per person admitted to the hospital did not suffer any alterations [53].

The second intervention, "Allow patient mobility between regions" [26], had the effect of increasing the efficiency of a provincial health organization, i.e. patients coming from other provinces in Italy, inside and outside the region, as they used the inputs that would otherwise be underemployed.

"Freedom of user choice between public and private providers" is the third intervention of this dimension, and for authors [26], users should have full freedom of choice on whether to be served by public or private service.

The fourth intervention was called "National Reimbursement System considering the regional characteristics of the population and structure of the health system". The intervention has a significant negative impact on health efficiency if it does not take into account regional characteristics of the population and structure of the health system [26].

The three outcomes of these interventions were rated "moderate recommendation" as they did not show a factor that could have increased their degree of recommendation, the dose-response gradient. Coming from the same study, its quality assessment was 8/8 (Table 2).

The fifth intervention was called "fiscal decentralization" and refers to the fiscal autonomy of regional health authorities (RHA). It is a financing mechanism, a quasi-market model with purchaser-provider. This model allows RHA to be responsible for delivering a basic package of health care services through a network of population-based local health authorities as well as public and private accredited hospital providers. A study conducted a convergence analysis to assess whether fiscal decentralization could improve two selected health indicators: the infant mortality rate (IMR) and the increase in life expectancy at birth (LEB). The authors claim that there is no clear indication of σ convergence or long-term σ divergence for both outcomes in patients' health. According to the authors, the effects of fiscal decentralization do not seem to demonstrate a systematic dynamic. For both outcomes in patients' health, all axis exhibits a negative slope (always statistically significant at 1% level), which is consistent with the hypothesis of β-convergence. Furthermore, as the level of decentralization increases, the slope of the axis (β coefficient) increases. This indicates the beneficial effect of higher degrees of fiscal decentralization in improving the convergence process. However, the fact that the axis intersect in the analysis indicates that the decentralization growth ratio is sensitive (moderate) to the level of health outcomes of the geographical region being analyzed. As the IMR decreases and LEB increases, the fiscal decentralization's efficiency in contributing to the reducing of the IMR or the increase of LEB is reduced [21].

The last intervention of this dimension was called "Vertical centralized programs in decentralized context" and was the only identified intervention with a negative effect. According to the authors [19], in a decentralized context, an ideal situation would be to have horizontal programs, as vertical centralized programs meet the decentralization proposal, impacting on their effectiveness. In the authors' study, there was a significant decline in DPT (diphtheria, pertussis and tetanus) vaccination coverage, from 82% in 1995, without service decentralization, to 63% in 1998, following the implementation of decentralization. The study was rated with 4/8, as it does not identify confounding factors that may interfere with the results, and did not perform statistical analyses. Thus, the outcome was rated as "very low recommendation".

The "rationalization" dimension included four interventions, "Provision of sanitary transport services between areas further from the polo hospital", "Concentration of procedures in high-volume hospitals", "Transfer of patients who arrived in low-volume hospitals to high-volume hospitals" and "Concentration of procedures in specialized hospitals". The first is important in regionalized contexts, as it ensures transportation to the patient whenever necessary. In a study identified in the literature [23], whose assessment was 8/8, it is preferable to opt for the transportation of emergency medical services of 100% of patients to hospitals with percutaneous coronary intervention rather than building new hospitals, with a cost of U$ 506/QALY (95% CI $474-$519). The outcome of the intervention was assessed as "moderate recommendation".

The intervention: "Concentration of procedures in high-volume hospitals" is defined as a minimum number of procedures in a given specialty that ensures better results at the lowest cost in health, given the distances between the referral hospital and the patient's residence. In all, 20 studies reported this intervention [20, 27–31, 33–35, 37–39, 41–43, 46, 47, 50, 52, 55, 56]. The studies showed a positive outcome regarding this intervention, reducing the days of

hospitalization and the in-hospital mortality rates, mainly. However, it is worth noting that this intervention has as limitation the distance between the hospital and the patient's residence; there is a maximum distance between them. The studies that showed this intervention received 8/8 evaluation, except for the studies by Lau et al. [39] and Grumbach et al. [33], which received 6/8 evaluation each, Imperato et al. [35], which received 7/8 evaluation, Tanke & Ikkersheim [50], who received "unclear" in two evaluation items, with a 6/8 evaluation, and the study by Hamilton et al. [34], which a 4/8 evaluation. However, both intervention outcomes were assessed as highly recommended since they showed the three factors that increase confidence in the results.

The intervention "Transferring patients who arrived from low-volume hospitals to high-volume hospitals" was identified in one study [18] and addresses the removal of patients who were admitted from low-volume hospitals to high-volume hospitals. For the authors, if the patient was not transferred, the expected quality-adjusted life years (QALYs) are 5.64 per patient, and the cost is $ 100,457.00. If the patient is transferred to a high-volume hospital, the expected QALYs are 7.23 per patient, and the cost is $ 117,284.00 per patient. Thus, regionalization results in a net gain of 1.60 QALYs per patient at a cost of $ 10,548.00/QALY. This study was evaluated as 8/8, while the outcome of its intervention received a "moderate recommendation" evaluation.

The intervention "Concentration of procedures in specialized hospitals", the last of the rationalization dimension, differs from high-volume hospitals in that they do not necessarily have a large number of medical procedures. These hospitals are classified according to their specialization: general, semi-specialized or specialized. One study [54] evaluated the 5-year survival rate in general, semi-specialized and specialized hospitals, with their values being 38.0 (95% CI 36.0–39,); 39.4 (95% CI 37.5–41.4) and 40.3 (95% CI 37.4–43.1), respectively. The study in question received 8/8 evaluation, while the outcome of the intervention was evaluated as "moderate recommendation".

Finally, only one intervention was found in the "governance" dimension, namely "Public Disclosure", which can be understood as the government's ability to publicize its data in order to improve the Government's credibility with the population. In a study found in the literature [48], the "public transparency ranking" governance model brought the most results to the Italian health system, and consolidated the systematic involvement of clinics and process improvement, supporting the identification of best practices and equal revision of mechanisms. The study received 4/8 evaluation for not identifying possible confounding factors nor performing statistical analysis. In turn, the outcome of the intervention was assessed as "very low recommendation", since in addition to several methodological problems it did not show any factor that could increase results reliability.

## Discussion

Although regionalization has been implemented for a long time, and in several countries, there are few studies analyzing the results of interventions to improve health indicators. In all, 16 interventions were identified that indicated different degrees of recommendations for improving the management of the health system regionalization. The results of the studies categorized as having a highly recommended intervention showed that regionalization was effective under administrative decentralization and for rationalization of resources. The most investigated intervention was the strategy of concentrating procedures in high-volume hospitals, which showed positive outcomes, especially with the reduction of hospitalization days and in-hospital mortality rates [20, 27–31, 33–35, 37–39, 41–43, 46, 47, 50, 52, 55, 56]. Other systematic reviews and primary studies that looked at different lines of care found similar results

regarding these outcomes [1, 59–64]. The gain in scale, higher learning curve of health professionals and greater bargaining power for the acquisition of inputs are among the favorable factors for the concentration of procedures in high-volume hospitals [65–67].

The reduction in mortality rate was the most commonly reported result in the studies analyzed. Scrutiny showed that regionalization is associated with reduced trauma-related mortality rates [59]. Regionalized trauma care systems allow to reduce delays in medical care, prevent inappropriate treatment, and especially reduce preventable deaths [68]. In addition, successful trauma care is largely time sensitive. The golden hour model is based on the idea that the quality and appropriateness of treatment in the first hour of care influences the patient's prognosis and is the basis for trauma regionalization [68]. Therefore, with an important ethical requirement, a regionalized care system ensures better care, equitably, to a large portion of patients with the best overall results.

It was also identified that the hospital volume of a given specialist guarantees better results with the lowest cost in health [18]. Thus, regionalization results in a net gain of 1.60 QALYs per patient at a cost of U\$ 10,548.00/QALY. Similarly, a systematic review identified that high-volume hospitals can reduce surgery costs. The authors indicated that minimally invasive surgeries for radical prostatectomy cost approximately U\$ 41,000 in low-volume hospitals, while high-volume surgeries cost approximately U\$ 28,000 [69].

Although our systematic review did not identify studies that investigated associated strategies (use of more than one intervention at a time), other systematic reviews have shown that associated interventions may be more effective [70–72]. Taking high-volume hospitals as an example, they are expected to be situated in environments with other interventions identified in this systematic review, such as: i) integration and coordination of care networks to ensure care at all levels of complexity (primary, secondary and tertiary healthcare); ii) outreach services to remote areas and sanitary transport to ensure access to healthcare for those distant from a high-volume hospital; iii) dissemination of data on health services so that the patient has information on the quality of health services; and iv) freedom of choice for the patient to select the health service they prefer.

Positive results were also observed in the dimension of decentralization of services, increasing the efficiency of the health system. In addition to increasing the proportion of the population served, actions related to the decentralization dimension can reduce inequality in health spending [17, 19, 26, 49, 53]. The literature also indicates that the decentralization of resources in the context of regionalization increases public participation in decision-making, as it brings local resource planning closer to the needs of the population [73].

User freedom of choice between public and private providers was also considered a relevant strategy for accessing the necessary services [26]. However, examples of users' lack of freedom to choose the professional who will perform their care in the literature are not rare [74–76]. In the logic of regionalized services, it seems to be even more difficult for a user to be free to choose this professional, as some of them will not be tied to the municipality they reside. This time, discussing the freedom to choose private contracted services seems immature compared to the discussions we still need to have, despite the potential gain that the intervention could bring to users.

Despite the positive results of regionalization, Cavalieri & Ferrante (2020) warn that they have not identified any obvious connections between fiscal decentralization and improvements in a population's health. This result is in agreement with the published literature on the subject. Di Novi et al. (2019) observed that, although fiscal decentralization helps to contain disparities between regions, there is no statistical difference in the analyzed models. The results indicate that wealthier health regions tend to obtain better results with fiscal decentralization, indicating the reduction of inequities in health. Underprivileged regions, however, continue to

rely on subsidy at central level, which does not render any effect for the RHA in terms of accountability and governance [77].

Moreover, the study indicates that the significance of the impact of fiscal decentralization increases over the years [77], supporting the idea that it takes time to observe the effects of regionalization on a population's health. This finding corroborates the need for further studies on health regionalization, since the topic still has critical gaps that have not yet been exhausted by its scientific literature.

## Limitations of the study

This review has some limitations. Firstly, there is no conceptual consensus on regionalization. The included primary studies defined regionalization differently, contributing to the heterogeneity of the identified results. Secondly, there are no well-established descriptors in the literature for "regionalization," so we may have 'missed' some relevant studies. In order to reduce this limitation, we used the McMaster Health Forum's Taxonomy of governance, financial and delivery arrangements, and implementation strategies within health systems [78] to support the search.

Thirdly, there is the methodological limitation of the included primary studies. The variation of the results did not allow to perform meta-analysis for the identified outcomes. In addition, the absence of controlled trials is emphasized, and studies with methodological designs of lower degree of evidence had to be included, most of them observational studies. Thus, it was not possible to isolate the effects of interventions. Potential effects of converging factors may have distorted the effect of the main intervention.

## Implications for politics

The first point to consider is that, although literature points to the possible effectiveness of regionalization, it must be considered that each country has its political, economic, and geographical structure, which makes it daring to propose a standard recipe for the implementation of health regionalization [79]. It's required that decision-makers, organized civil and academic society to evaluate the necessary changes and feasibility for the implementation of an evidence-informed policy, taking in consideration the uncertainties of the evidences and balancing different results [80].

It is necessary to consider that the implementation of regionalization involves changes in current standards of practice and in the distribution of health resources, especially specialized services. Services need to be regionalized based on geographic organization, so that different entities can organize and manage the provision of services and programs [73]. However, flexibility in patient redistribution is necessary because transferring patients from low-volume centers to high-volume centers may improve outcomes in some conditions but worsen them for others. For Lumpkin & Stitzenberg, regionalization may influence access to care for vulnerable populations by increasing the barriers to allow for care for many patients due to distances between services [81].

In addition, other regionalization strategies were identified. Due to the lack of a single definition for these outcomes and the small number of studies reporting them, there is a need for further studies on the resulting effects. However, it should be noted that integrating services in a coordinated manner broadens the range of services provided and has the potential to reduce health costs.

The economic impact of centralizing cancer services will likely vary depending on many factors, such as tumor type, treatment selected, and geographic location. Costs may be reduced by increasing the surgeon's volume, but it is not clear from the current evidence base what the optimal volume would be. Even if centralizing cancer services results in cost savings for

healthcare providers and patients (in terms of better health outcomes), this can be offset by increased patient access costs in terms of increased travel time and distance [82]. Several studies have shown that increasing the distance and travel time from patients' homes to centralized cancer services reduces the likelihood of treatment adherence and acceptance and thus contributes to increased health inequality. Given all these issues, it is likely that there is no "one size fits all" centralization model to address all cancers, treatment modalities, and sites.

## Implications for research

After verifying the studies included in this review, a lack was identified in primary studies. The absence of randomized controlled trials is the first barrier that must be overcome, thus corroborating to more accurately estimating the effects of regionalization. In addition, most of the included studies were related to high per capita income countries, and it is necessary to investigate these interventions in low- and middle-income countries.

The identified outcomes had little epidemiological results for users in general, as they were limited to hospitalized cases (mortality and length of stay). Governance and service arrangements have been poorly evaluated in the identified primary studies. Moreover, the vast majority of investigated interventions come from only one study, which limits inference to other contexts.

## Supporting information

**S1 Appendix. Attachment A—full-text articles excluded list, with reasons.**
(DOCX)

**S2 Appendix. Attachment B—characteristic of studies included.**
(XLSX)

**S1 Checklist. PRISMA 2009 checklist.**
(DOC)

## Author Contributions

**Conceptualization:** Maíra Catharina Ramos, Jorge Otávio Maia Barreto, Helena Eri Shimizu, Everton Nunes da Silva.

**Data curation:** Maíra Catharina Ramos, Jorge Otávio Maia Barreto, Helena Eri Shimizu, Everton Nunes da Silva.

**Formal analysis:** Maíra Catharina Ramos, Jorge Otávio Maia Barreto, Helena Eri Shimizu, Everton Nunes da Silva.

**Funding acquisition:** Helena Eri Shimizu.

**Investigation:** Maíra Catharina Ramos, Everton Nunes da Silva.

**Methodology:** Maíra Catharina Ramos, Amanda Pereira Gomes de Moraes, Everton Nunes da Silva.

**Project administration:** Maíra Catharina Ramos.

**Resources:** Maíra Catharina Ramos, Amanda Pereira Gomes de Moraes, Everton Nunes da Silva.

**Software:** Amanda Pereira Gomes de Moraes.

**Supervision:** Everton Nunes da Silva.

**Validation:** Maíra Catharina Ramos, Everton Nunes da Silva.

**Writing – original draft:** Maíra Catharina Ramos, Helena Eri Shimizu, Everton Nunes da Silva.

**Writing – review & editing:** Maíra Catharina Ramos, Helena Eri Shimizu, Everton Nunes da Silva.

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
