## [Decision Letter · Decision Letter 0]

14 Feb 2020

PONE-D-19-27817

Regionalization for health improvement: a systematic review

PLOS ONE

Dear Mrs Ramos,

Thank you for submitting your manuscript to PLOS ONE. After careful consideration, we feel that it has merit but does not fully meet PLOS ONE’s publication criteria as it currently stands. Therefore, we invite you to submit a revised version of the manuscript that addresses the points raised during the review process.

The manuscript has been assessed by two reviewers. Their comments are appended below. The reviewers have raised significant concerns about the manuscript.

We would appreciate receiving your revised manuscript by Mar 30 2020 11:59PM. Please include the following items when submitting your revised manuscript:

We look forward to receiving your revised manuscript.

Kind regards,

Arianne Elissen, Ph.D.

Academic Editor

PLOS ONE

Journal Requirements:

- Ke, K. M., W. Hollingworth, and A. R. Ness. "The costs of centralisation: a systematic review of the economic impact of the centralisation of cancer services." European journal of cancer care 21.2 (2012): 158-168.

The text that needs to be addressed involves a paragraph of the Discussion.

In your revision ensure you cite all your sources (including your own works), and quote or rephrase any duplicated text outside the methods section. Further consideration is dependent on these concerns being addressed.

3. a)    Please provide an amended Funding Statement that declares *all* the funding or sources of support received during this specific study (whether external or internal to your organization) as detailed online in our guide for authors at http://journals.plos.org/plosone/s/submit-now.  

b)    Please state what role the funders took in the study.  If any authors received a salary from any of your funders, please state which authors and which funder. If the funders had no role, please state: "The funders had no role in study design, data collection and analysis, decision to publish, or preparation of the manuscript."

5. Please include a copy of Table 33 which you refer to in your text on page 16.

Reviewers' comments:

Reviewer's Responses to Questions

**Comments to the Author**

1. Is the manuscript technically sound, and do the data support the conclusions?

Reviewer #1: Partly

Reviewer #2: Partly

2. Has the statistical analysis been performed appropriately and rigorously? 

Reviewer #1: No

Reviewer #2: Yes

3. Have the authors made all data underlying the findings in their manuscript fully available?

Reviewer #1: Yes

Reviewer #2: Yes

4. Is the manuscript presented in an intelligible fashion and written in standard English?

Reviewer #1: No

Reviewer #2: No

5. Review Comments to the Author

Reviewer #1: The article entitled, “Regionalization for health improvement: a systematic review” presents an ambitious research question, aiming to summarize interventions that have been successful at the organization of health care services in order to guide future recommendations. To answer this question, the authors have chosen a literature review to find original research describing the impact of such reorganization policies. The results summarize the articles found in this search and conclusions remain broad as the study results are quite varied.

There are a few overall themes which need to be better addressed by the authors in order to meet the aims they have set out in this study.

1. The definition of regionalization used in this research does not appear to match more common definitions. Traditionally, regionalization refers to the centralization of health services in one location – such as high volume or specialized centers. The authors use of the term seems to be more in the vein of organization of health services. This warrants renaming the study and substituting the term regionalization for “optimal organization” or some alternative phrasing.

2. The methodology needs to be revised as detailed in comments below.

3. The organization of the results with use of the terms regionalization, rationalization, and decentralization need to be presented differently.

4. I would recommend focusing on either health outcomes or distribution of resources instead of the effects of health system reorganization on both

Abstract:

Page 2 – The first sentence of the abstract is unclear to me and does not appear to match the aims of the study. “The health services regionalization” may be better worded differently. Perhaps the intent is to describe interventions aimed at the regionalization of health services and describe their impact. The second sentence is somewhat repetitive of the first and similarly should be reworded. The health and management indicators mentioned should also be more clearly defined further in the study. Further in the abstract it states “the health regionalization system” which is also an unclear term and should be better defined. I would recommend defining regionalization in the abstract.

“The results of the studies included as a highly recommended intervention showed that

regionalization was effective within the decentralization and rationalization of resources” – this sentence should be made more clear and again the terms regionalization, decentralization, and rationalization should be more clearly defined and do not appear to be used effectively here.

“The most investigated intervention was the adoption of the strategy of concentration of

procedures in high-volume hospitals,” this is more commonly the definition of regionalization.

Introduction:

Page 2 – please re-evaluate the definition of regionalization used in the paper.

Page 3 – The first paragraph starts by defining decentralization as part of regionalization, whereas more commonly they are considered to be opposing ways of organization health services.

Page 3 – again would not consider decentralization as part of regionalization. This should instead be changed to describe the organization or distribution of health services.

Page 3, last paragraph – this is not complete and again goes back to the comment on the abstract on how this is not a clear statement. “The aim of this study is to identify and characterize effective interventions to

improve health and management indicators within the health services regionalization. A

systematic review was conducted to answer the following question:”

Methodology:

Page 4 – please redefine the question.

Page 4 – Outcomes – the health management outcomes need some basis for where these came from. Ambulatory care-sensitive conditions should be defined.

Page 4 – Search strategy – the search string should instead include OR between these brackets instead of AND to cast a wider net, at least between the two first strings of searches. Especially considering the regionalization and decentralization terms used. There are also additional synonymous terms which should be used which have been omitted – including devolution which is used in place of decentralization and centralization which is synonymous with regionalization. Here is an example of a more broad search for regionalization and decentralization and I would recommend all these terms be used at a minimum.

Decentralization/Regionalization Search String (https://www.ncbi.nlm.nih.gov/pubmed/31657175)

(district hospital) OR (community health center) OR (primary health centre) OR (primary health center) OR (community facility) OR rural OR decentralization OR decentralized OR decentralized OR regionalization OR regionalized OR regionalised OR (First-level hospital) OR (referral hospital) OR (specialised hospital) OR (specialized hospital) OR (devolution) OR (devolved) OR (devolve) OR (devolvement) OR (devolving) OR (centralization) OR (centralized) OR (centralised)

The last search string on outcomes should include mortality and health outcomes more broadly.

Page 8 – please define the GRADE system in 1 or 2 sentences and it’s application to this study more clearly.

Results

Page 9 – The first two paragraphs are repetitive of the first table and can be shortened.

Page 9 – The paragraph on the GRADE system is descriptive of methods and should be included there

Table 3 – the dimension and interventions chosen here need more of a basis. Either in the introduction or methods – there should be a better description of how these dimensions were settled on.

- Define the “polo hospital” in table 3

- The rationalization term used here is synonymous with regionalization broadly in the literature

Discussion

Page 19 - “The results of the studies included as a highly recommended intervention showed that

regionalization was effective within the decentralization and rationalization of resources.” Can this be differently worded? Unclear of authors’ intention with this statement.

Because the results of the articles are so varied, there is really only one reasonable conclusion that can be drawn, as the authors point out – the concentration of procedures in high volume hospitals has shown favorable health outcomes.

This has been shown in previous studies, as the authors point out, so it is unclear what new perspective this study brings to the literature.

Other limitations of the study include the literature is largely from the USA and HICs. There are only two low income countries included. The authors mention this in the last two paragraphs. Recommendations or insights are therefore only attributable to HICs. I would appreciate further discussion on the disparities in different health settings and why or why not such interventions recommended in this study may be successful in these areas.

Because studies on this scale are hard to conduct, modeling studies are sometimes more appropriate to answer these health organization questions (https://www.thelancet.com/journals/langlo/article/PIIS2214-109X(19)30170-6/fulltext)

Reviewer #2: The manuscript presents originality in its proposal and important relevance for health services regionalization.

The authors were able to organize a methodological strategy that allowed them to argue that the difficulties of articulating health services regionalization in differents aspects resulted from dimmensions. Based on these results, they were able to elaborate a set of recommendations that can mitigate the articulation difficulties identified.

I suggest the authors make clear the regionalization defines they are using in the manuscript. I recommend indicating the countries are using health regionalization and to indicate their success. Review if is correct in the end of introduction, it is looks like are missing some questions.

I recommend use the standard Plos one Reviewer Guidelines.

The discussion of results follows the logic of the entire study, that is, an investigation about different realitys, certainly with few similarities in other latitudes. However, this does not prevent to identify aspects overlapping with the reality of other health systems.

I recommend a major revision of the English grammar, as this would make the text more comprehensible to all readers.

6. PLOS authors have the option to publish the peer review history of their article (what does this mean?). If published, this will include your full peer review and any attached files.

Reviewer #1: No

Reviewer #2: No

---

## [Author Response · Author response to Decision Letter 0]

13 Aug 2020

Dear Editor,

We are thankful for the contributions to our manuscript “Regionalization for health improvement: a systematic review”. This evaluation is of utmost importance to help us improve and enrich our paper. We present the responses addressing each of the questions below.

Best regards, 

The authors

Comment 1, Editor: “We noticed you have some minor occurrence of overlapping text with the following previous publication(s), which needs to be addressed: Ke, K. M., W. Hollingworth, and A. R. Ness. "The costs of centralisation: a systematic review of the economic impact of the centralisation of cancer services." European journal of cancer care 21.2 (2012): 158-168. The text that needs to be addressed involves a paragraph of the Discussion. In your revision ensure you cite all your sources (including your own works), and quote or rephrase any duplicated text outside the methods section. Further consideration is dependent on these concerns being addressed”.

Our response: We are thankful for the time spent on correcting our manuscript and the contributions mentioned. The paragraph in question was rewritten and properly cited.

Comment 2, Editor: “Please provide an amended Funding Statement that declares *all* the funding or sources of support received during this specific study (whether external or internal to your organization) as detailed online in our guide for authors at http://journals.plos.org/plosone/s/submit-now”. 

Our response: The authors received no specific funding for this work.

Comment 3, Editor: “Please state what role the funders took in the study. If any authors received a salary from any of your funders, please state which authors and which funder. If the funders had no role, please state: "The funders had no role in study design, data collection and analysis, decision to publish, or preparation of the manuscript."

Our response: The authors received no specific funding for this work.

Comment 4, Editor: “Please complete your Competing Interests on the online submission form to state any Competing Interests. If you have no competing interests, please state "The authors have declared that no competing interests exist.", as detailed online in our guide for authors at http://journals.plos.org/plosone/s/submit-now

 This information should be included in your cover letter; we will change the online submission form on your behalf.”

Our response: We included the sentence "The authors have declared no conflict of interest.", as requested.

Comment 5, Editor: “Please include a copy of Table 33 which you refer to in your text on page 16”.

Our response: We believe there was a typing mistake, as the correct Table should be number 3. Table 3 is located in page 13. 

Comment 6, Editor: “Please include captions for your Supporting Information files at the end of your manuscript, and update any in-text citations to match accordingly. Please see our Supporting Information guidelines for more information: http://journals.plos.org/plosone/s/supporting-information.”

Our response: The captions were included and the citations were updated.

Comment 1, reviewer 1: “The article entitled, “Regionalization for health improvement: a systematic review” presents an ambitious research question, aiming to summarize interventions that have been successful at the organization of health care services in order to guide future recommendations. To answer this question, the authors have chosen a literature review to find original research describing the impact of such reorganization policies. The results summarize the articles found in this search and conclusions remain broad as the study results are quite varied. There are a few overall themes which need to be better addressed by the authors in order to meet the aims they have set out in this study.

1. The definition of regionalization used in this research does not appear to match more common definitions. Traditionally, regionalization refers to the centralization of health services in one location – such as high volume or specialized centers. The authors use of the term seems to be more in the vein of organization of health services. This warrants renaming the study and substituting the term regionalization for “optimal organization” or some alternative phrasing.”

Our response: We are thankful for the time spent on correcting our manuscript and the contributions mentioned. We have understood the argument of the reviewer and consider it interesting, however the subject of regionalization is still highly debated, and as such there is little consensus on the subject as a public policy. Certain public health systems have decided to regionalize their services. Canada and Brazil are prominent examples of countries that adopted the policy of regionalization in health as a means to organize healthcare services (Marchildon, 2019). Whenever the literature indicates that one of the characteristics of regionalization is decentralization, it is referring to “to decentralize responsibility for the administration of major health programs and to centralize authority over the operation of these same programs by eliminating hospital boards and other local health agencies” (Baker et al, 2017). According to Reamy (1995), decentralization is an important subject to regionalization. It implies in the transference of any degree of authority and responsibility towards a lower level or region. In some cases, decentralization and regionalization occur simultaneously, as the region might be decentralized in terms of governance but centralized in terms of providing hospital services, as mentioned by the reviewer. It should also be noted that when characterizing regionalization as decentralized we are referring to reorganization at all healthcare levels (primary, secondary and tertiary care), not just a specific kind or level of care such as hospital care. Moreover, Brazil, Canada and Italy, for instance, have adopted the policy of regionalizing their systems of healthcare services together with the reorganization of healthcare networks – also known in other countries as integrated service networks, regional networks, integrated services, and integrated care (Viana et al, 2018). Healthcare networks provide a new logic for the organization of health services, defined as a set of actions and services articulated at increasing levels of complexity in order to ensure the integrality of healthcare.

References: 

1. Marchildon G. The integration challenge in Canadian regionalization. Cadernos de Saúde Pública. 2019; 35 Sup 2:e00084418.

2. Barker, P., & Church, J. (2017). Revisiting Health Regionalization in Canada: More Bark Than Bite? International Journal of Health Services, 47(2), 333–351. https://doi.org/10.1177/0020731416681229).

3. Reamy J. Health Service Regionalization in New Brunswick, Canada: A Bold Move. International Journal of Health Services. 1995; 25(2): 271–282.

4. Viana Ana Luiza d’Ávila, Bousquat Aylene, Melo Guilherme Arantes, Negri Filho Armando De, Medina Maria Guadalupe. Regionalização e Redes de Saúde. Ciênc. saúde coletiva [Internet]. 2018 June; 23( 6 ): 1791-1798. 

Comment 2, reviewer 1: “2. The methodology needs to be revised as detailed in comments below. 3. The organization of the results with use of the terms regionalization, rationalization, and decentralization need to be presented differently.”

Our response: As mentioned in the previous item, the results were presented according to the concept utilized by authors studying regionalization, which in turn is the integrated organization of a healthcare system with several coordinated functions that serves a specific territory wherein a regional governance structure is responsible for providing and administrating health services of that specific region (Aerde, 2016). For this reason, we present the results in the dimensions: decentralization, rationalization, coordination, and governance. 

Reference: 

5. Aerde, JV. Has regionalization of the Canadian health system contributed to better health?. Canadian Journal of Physician Leadership, 2016; 2(3):65-70. 

Comment 3, reviewer 1: “4. I would recommend focusing on either health outcomes or distribution of resources instead of the effects of health system reorganization on both”.

Our response: We have understood this recommendation and are thankful for the suggestion, however, the choice for presenting the results of regionalization in the allocation of resources and in its efficacy of health outcomes was made due to the number of studies identified for both. Several studies demonstrate the importance of centralizing high complexity services and how this centralization is necessary to optimize resources for health; however, few are those that describe the results of adopting regionalization as a service organization policy. As such, we opted to keeping both results to present the evidence identified in the literature concerning the adoption of the regionalization policy in health in order to report the decision-making to administrators that might be interested in adopting a similar policy in their countries.

Comment 4, reviewer 1: “Abstract: Page 2 – The first sentence of the abstract is unclear to me and does not appear to match the aims of the study. “The health services regionalization” may be better worded differently. Perhaps the intent is to describe interventions aimed at the regionalization of health services and describe their impact. The second sentence is somewhat repetitive of the first and similarly should be reworded. The health and management indicators mentioned should also be more clearly defined further in the study. Further in the abstract it states “the health regionalization system” which is also an unclear term and should be better defined. I would recommend defining regionalization in the abstract. “The results of the studies included as a highly recommended intervention showed that regionalization was effective within the decentralization and rationalization of resources” – this sentence should be made more clear and again the terms regionalization, decentralization, and rationalization should be more clearly defined and do not appear to be used effectively here. “The most investigated intervention was the adoption of the strategy of concentration of procedures in high-volume hospitals,” this is more commonly the definition of regionalization”. 

Our response: We included the excerpt “Regionalization is the integrated organization of a healthcare system, wherein regional structures are responsible for providing and administrating health services in a specific region. This method was adopted by several countries to improve the quality of provided care and to properly utilize available resources” in the abstract. As for the use of the term decentralization, we have previously justified the reason for our choice.

Comment 5, reviewer 1: “Introduction: Page 2 – please re-evaluate the definition of regionalization used in the paper. Page 3 – The first paragraph starts by defining decentralization as part of regionalization, whereas more commonly they are considered to be opposing ways of organization health services. Page 3 – again would not consider decentralization as part of regionalization. This should instead be changed to describe the organization or distribution of health services.”

Our response: The definition of regionalization was included as understood by the authors. In order to address the reviewer’s request and clarify the utilized concept, we included the following excerpt:

“The literature shows the Dawson Report as one of the first and main documents to concern regionalization in 1920. The author pictured an integrated healthcare system wherein the prevention of diseases and healthcare promotion would be coordinated, including primary, secondary and tertiary care of both short and long term serving to organize health services in territories specifically selected for such services[3]. England, Canada and Brazil are amidst the countries that used Dawson’s ideals to develop health services regionalization policies to integrate the coordination of institutional and community integration in order for the community to participate, and to transfer the authority of management concerning healthcare programs to regional level. The centralization of governance by eliminating local healthcare entities and agencies in these countries was followed by the decentralization of responsibility towards providing most health services[4]. Likewise, there is no consensus on the concept of regionalization. In federative countries, the concept that is closest to regionalization is: the integrated organization of a healthcare system with several coordinated functions that serves a specific territory wherein a regional structure would be responsible to provide and administrate healthcare services[5,6]. 

 According to Marchildon[7], the goal of regionalization is to integrate and rationalize healthcare services, to promote evidence-based practice, to decentralize resources and decision-making, and to switch focus and resources towards the prevention of ilnesses and promotion of health”. 

We also reworded the last paragraph: 

“This study thus sought to identify and characterize effective interventions to improve health and management indicators within the scope of the regionalization of health services. A systematic review was conducted to answer the following question: what are effective interventions to improve health and management indicators within the health services regionalization?”

Comment 6, reviewer 1: “Page 3, last paragraph – this is not complete and again goes back to the comment on the abstract on how this is not a clear statement. “The aim of this study is to identify and characterize effective interventions to improve health and management indicators within the health services regionalization. A systematic review was conducted to answer the following question:”

Our response: Indeed, the last paragraph was incomplete. We have altered it to: 

“This study thus sought to identify and characterize effective interventions to improve health and management indicators within the scope of the regionalization of health services. A systematic review was conducted to answer the following question: what are effective interventions to improve health and management indicators within the health services regionalization?”.

Comment 7, reviewer 1: “Methodology: Page 4 – please redefine the question”.

Our response: As previously justified, we used the terms regionalization and decentralization in a joint manner and not as opposite concepts, as stressed by the reviewer, due to the concept of regionalization adopted in the public policies of countries such as Canada, Brazil and Italy. We agree that centralization of highly specialized care is a common and important strategy in several countries, however we are referring to political and administrative decentralization of the healthcare system, thus justifying the question and search utilized.

Comment 8: “Page 4 – Outcomes – the health management outcomes need some basis for where these came from. Ambulatory care-sensitive conditions should be defined.”

Our response: The following excerpt was included: 

“The Taxonomy of Decisions of McMaster University was used to define effects on management. This taxonomy allows for better standardization of health descriptors, as it proposes useful patterns for ideal decision-making and provides tools to take good practical decisions[15]”.

Comment 9, reviewer 1: “Page 4 – Search strategy – the search string should instead include OR between these brackets instead of AND to cast a wider net, at least between the two first strings of searches. Especially considering the regionalization and decentralization terms used. There are also additional synonymous terms which should be used which have been omitted – including devolution which is used in place of decentralization and centralization which is synonymous with regionalization. Here is an example of a more broad search for regionalization and decentralization and I would recommend all these terms be used at a minimum. Decentralization/Regionalization Search String (https://www.ncbi.nlm.nih.gov/pubmed/31657175) (district hospital) OR (community health center) OR (primary health centre) OR (primary health center) OR (community facility) OR rural OR decentralization OR decentralized OR decentralized OR regionalization OR regionalized OR regionalised OR (First-level hospital) OR (referral hospital) OR (specialised hospital) OR (specialized hospital) OR (devolution) OR (devolved) OR (devolve) OR (devolvement) OR (devolving) OR (centralization) OR (centralized) OR (centralised)”

Our response: As previously justified, “regionalization AND decentralization” was used due to the concepts of regionalization and decentralization utilized.

Comment 10, reviewer 1: “The last search string on outcomes should include mortality and health outcomes more broadly”.

Our response: The Taxonomy of decisions of the McMaster University was used for choosing the terms utilized in the search, as previously explained.

Comment 11, reviewer 1: “Page 8 – please define the GRADE system in 1 or 2 sentences and it’s application to this study more clearly.”

Our response: The paragraph concerning GRADE was altered to: 

“The Grading of Recommendations Assessment, Development and Evaluation (GRADE) was adopted to evaluate the results. The GRADE system is a transparent structure for developing and presenting evidence summaries, providing a systematic approach for recommendations to improve clinical practice, with clear and concise information over the quality of evidence (the degree to which a given result might be trusted) and the reliability of the recommendation[16]. With GRADE, the interventions identified during systematic review were stratified as “Level of recommendation: very low recommendation; low recommendation; moderate recommendation, and; high recommendation”.

Comment 12, reviewer 1: “Results. Page 9 – The first two paragraphs are repetitive of the first table and can be shortened.”

Our response: The first two paragraphs were revised as suggested by the reviewer.

Comment 13, reviewer 1: “Page 9 – The paragraph on the GRADE system is descriptive of methods and should be included there”. 

Our response: Information concerning GRADE was included in the methodology, as previously reported.

Comment 14, reviewer 1: “Table 3 – the dimension and interventions chosen here need more of a basis. Either in the introduction or methods – there should be a better description of how these dimensions were settled on”.

Our response: The dimensions were defined in the introduction, as follows:

“Studies show that regionalization needs organizational arrangements to achieve its proper functioning[9,10], namely: i) coordination: concerns the integration, sharing, articulation and decision-making based on norms, legal or not, whose actors agree to share decisions and tasks; in forums and political mechanisms for intergovernmental negotiation; the functioning of representative institutions; in the coordinating and/or inducing role of Brazilian government[11]; ii) decentralization: defined as the delegation or devolution of certain responsibilities and functions, broadening the view on the autonomy of other federative entities[12]; iii) rationalization: it is defined as a resource utilization tool aiming to increase the efficiency of services without bias towards the effectiveness of the actions[13]; iv) governance: it is defined as the actions and means adopted by society to organize itself to protect and promote the health of the population, seeking to harmonize the decisions and actions of different actors in favor of the equity and sustainability of health systems[14]”. 

Comment 15, reviewer 1: “Define the “polo hospital” in table 3”

Our response: We have changed the term “polo hospital” to “high-volume hospital”.

Comment 16, reviewer 1: “The rationalization term used here is synonymous with regionalization broadly in the literature”. 

Our response: Regionalization in health is understood as a policy to organize services in an integrated and coordinate manner, including political-administrative transference to specific territories and optimally allocating resources to meet health needs. As such, rationalization is an important concept under the scope of health regionalization, however, regionalization itself is broader than the effective utilization of health resources.

Comment 17, reviewer 1: “Discussion. Page 19 - “The results of the studies included as a highly recommended intervention showed that regionalization was effective within the decentralization and rationalization of resources.” Can this be differently worded? Unclear of authors’ intention with this statement.”

Our response: As previously justified, we presented the results in the dimensions decentralization, rationalization, coordination, and governance as they are key concepts for health regionalization.

Comment 18, reviewer 1: “Because the results of the articles are so varied, there is really only one reasonable conclusion that can be drawn, as the authors point out – the concentration of procedures in high volume hospitals has shown favorable health outcomes. This has been shown in previous studies, as the authors point out, so it is unclear what new perspective this study brings to the literature.”

Our response: Despite other results not including clear evidence, it is important to document and discuss them as possible results of the implementation of a regionalization policy in the country. It must be noted that in the field of public policies uncertainty or lack of evidence is not comparable to their inefficacy. Unfortunately, we still find it difficult to document public policy cases implemented in certain countries to inform the local experience of other countries. However, evidencing actions with potential effect on public health contributes to the reduction of the know-do gap and helps the decision-making process in health.

Comment 19, reviewer 1: “Other limitations of the study include the literature is largely from the USA and HICs. There are only two low income countries included. The authors mention this in the last two paragraphs. Recommendations or insights are therefore only attributable to HICs. I would appreciate further discussion on the disparities in different health settings and why or why not such interventions recommended in this study may be successful in these areas.”

Our response: The following excerpt was included:

“Implications for politics

 The first point to consider is that, although literature points to the possible effectiveness of regionalization, it must be considered that each country has its political, economic, and geographical structure, which makes it daring to propose a standard recipe for the implementation of health regionalization[73]. It's required that decision-makers, organized civil and academic society to evaluate the necessary changes and feasibility for the implementation of an evidence-informed policy, taking in consideration the uncertainties of the evidences and balancing different results[74].”

 

Comment 1, reviewer 2: “The manuscript presents originality in its proposal and important relevance for health services regionalization. The authors were able to organize a methodological strategy that allowed them to argue that the difficulties of articulating health services regionalization in differents aspects resulted from dimmensions. Based on these results, they were able to elaborate a set of recommendations that can mitigate the articulation difficulties identified. I suggest the authors make clear the regionalization defines they are using in the manuscript. I recommend indicating the countries are using health regionalization and to indicate their success.” 

Our response: We are thankful for the time spent on correcting our manuscript and the contributions mentioned. We have included the definition of regionalization used in the study, as you can see in Comment 5, reviewer 1.

Comment 2, reviewer 2: “Review if is correct in the end of introduction, it is looks like are missing some questions. I recommend use the standard Plos one Reviewer Guidelines.”

Our response: Following the reviewer’s observation, we have changed the last paragraph to: 

“This study thus sought to identify and characterize effective interventions to improve health and management indicators within the scope of the regionalization of health services. A systematic review was conducted to answer the following question: what are effective interventions to improve health and management indicators within the health services regionalization?

Comment 3, reviewer 2: “The discussion of results follows the logic of the entire study, that is, an investigation about different realitys, certainly with few similarities in other latitudes. However, this does not prevent to identify aspects overlapping with the reality of other health systems.”

Our response: The following excerpt was included:

“Implications for politics

 The first point to consider is that, although literature points to the possible effectiveness of regionalization, it must be considered that each country has its political, economic, and geographical structure, which makes it daring to propose a standard recipe for the implementation of health regionalization[73]. It's required that decision-makers, organized civil and academic society to evaluate the necessary changes and feasibility for the implementation of an evidence-informed policy, taking in consideration the uncertainties of the evidences and balancing different results[74].”

Comment 4, reviewer 2: “I recommend a major revision of the English grammar, as this would make the text more comprehensible to all readers.”

Our response: We have contracted a certified language services provider to revise this paper and make it easier to understand for all readers.

---

## [Decision Letter · Decision Letter 1]

14 Oct 2020

PONE-D-19-27817R1

Regionalization for health improvement: a systematic review

PLOS ONE

Dear Dr. Ramos,

Thank you for submitting a revision for your manuscript to PLOS ONE. Attached below are comments made by one reviewer following evaluation of your revised manuscript.  I invite you to submit a new revised version of the manuscript that addresses the points raised during the review process.

We look forward to receiving your revised manuscript.

Kind regards,

Itamar Ashkenazi

Academic Editor

PLOS ONE

Reviewers' comments:

Reviewer's Responses to Questions

**Comments to the Author**

1. If the authors have adequately addressed your comments raised in a previous round of review and you feel that this manuscript is now acceptable for publication, you may indicate that here to bypass the “Comments to the Author” section, enter your conflict of interest statement in the “Confidential to Editor” section, and submit your "Accept" recommendation.

Reviewer #1: (No Response)

2. Is the manuscript technically sound, and do the data support the conclusions?

Reviewer #1: Partly

3. Has the statistical analysis been performed appropriately and rigorously? 

Reviewer #1: No

4. Have the authors made all data underlying the findings in their manuscript fully available?

Reviewer #1: Yes

5. Is the manuscript presented in an intelligible fashion and written in standard English?

Reviewer #1: Yes

6. Review Comments to the Author

Reviewer #1: The authors have made substantial improvements to the initial manuscript and I sincerely appreciate their consideration and response to my prior comments. However, the study still has some significant methodologic flaws. While I appreciate the authors comments clarifying the concepts of regionalization and decentralization being used in this study, their search strategy for studies does not match the main objectives. One of my prior comments was to expand their search strategy as to not miss out on publications that may mention regionalization, the main focus of this current study, in isolation of the other terms that are mentioned in the search strategy: decentralization, rationalization. I would still strongly recommend revising the search strategy to more broadly search for regionalization in accordance to my prior comment. One way to limit the results further would be to use the regionalization terms then include a longer AND statement to get to the components of health structuring they are hoping to capture further: for instance regionalization (including all relevant terms) AND (decentralization terms OR coordination terms OR rationalization terms OR governance OR outcome terms). The current search strategy is severely limiting to papers that include regionalization AND decentralization AND organizational strategy/outcomes. It also appears several of the search terms would need to be expanded to include all relevant permutations. For example, "Mortality rate" OR "child mortality rate" are included, but not mortality and not appropriate MeSH terms for these.

Accordingly, they should also acknowledge adherence to the research guidelines for systematic reviews, PRISMA, as in http://www.prisma-statement.org/.

One small comment regards the research question, for which I would recommend a slight rewording. "A systematic review was conducted to answer the following question:

what are effective interventions to improve health and management indicators within the

health services regionalization?"

I recommend striking the word THE and instead it should read: "what are effective interventions to improve health and management indicators within

health services regionalization?"

I look forward to seeing the updates on this study after the revision of the search strategy.

7. PLOS authors have the option to publish the peer review history of their article (what does this mean?). If published, this will include your full peer review and any attached files.

Reviewer #1: No

---

## [Author Response · Author response to Decision Letter 1]

1 Dec 2020

Dear Reviewer,

We thank you for the opportunity to have our manuscript PONE-D-20-14125R1, entitled "Regionalization for health improvement: a systematic review", evaluated for publication by Plos One. We also would like to thank the reviewer for his/her consideration and time spent reviewing our manuscript.

Yours sincerely, 

The Authors

Reviewer 1, comment 1: “The authors have made substantial improvements to the initial manuscript and I sincerely appreciate their consideration and response to my prior comments. However, the study still has some significant methodologic flaws.

While I appreciate the authors comments clarifying the concepts of regionalization and decentralization being used in this study, their search strategy for studies does not match the main objectives. One of my prior comments was to expand their search strategy as to not miss out on publications that may mention regionalization, the main focus of this current study, in isolation of the other terms that are mentioned in the search strategy: decentralization, rationalization. I would still strongly recommend revising the search strategy to more broadly search for regionalization in accordance to my prior comment. One way to limit the results further would be to use the regionalization terms then include a longer AND statement to get to the components of health structuring they are hoping to capture further: for instance regionalization (including all relevant terms) AND (decentralization terms OR coordination terms OR rationalization terms OR governance OR outcome terms). The current search strategy is severely limiting to papers that include regionalization AND decentralization AND organizational strategy/outcomes. It also appears several of the search terms would need to be expanded to include all relevant permutations. For example, "Mortality rate" OR "child mortality rate" are included, but not mortality and not appropriate MeSH terms for these.”

Our response: Thank you for taking the time to review our manuscript and for the helpful comments you have provided.

As it is a complex issue, we had previously included outcomes in the search strategy following the guide to a systematic review of the University of York, Chapter 3 - Systematic reviews of public health interventions. Understanding the reviewer's questioning, we redid the search strategy accordingly, excluding the outcomes. This change caused an increase in the articles identified from 657 to 3918. With the new search suggested, the strategy was used:

#1 MeSH descriptor: [Regional Health Planning]

#2 (regional governments):ti,ab,kw

#3 (regional healthcare):ti,ab,kw 

#4 (areawide planning):ti,ab,kw

#5 (regional healthcare network):ti,ab,kw

#6 (comprehensive health planning):ti,ab,kw

#7 (annual implementation plans):ti,ab,kw

#8 (regional governments):ti,ab,kw

#9 (regionalization):ti,ab,kw

#10 OR{#1-#9} or #2 or #3 or #4 or #5 or #6 or #7 or #8 or #9

#11 (decentralization):ti,ab,kw

#12 (coordination):ti,ab,kw

#13 (co ordination):ti,ab,kw

#14 (rationalization):ti,ab,kw

#15 (governance):ti,ab,kw

#16 #11 or #12 or #13 or #14 or #15

#17 #10 and #16

Except for 3 studies, the studies previously included were identified in a new search. The selection by title and abstract was carried out in a paired and independent manner by four researchers, to comply with the established deadline. In total, 119 studies were selected for a full reading. The inclusion stage was also carried out in a paired and independent manner by two researchers, with 39 meetings the inclusion criteria established, that is, 6 more than the previous search. Because of the change in the search, we inform that the METHODOLOGY section of the article has also undergone changes, which are highlighted below:

METHODOLOGY

The following search strategy was used: ("regional health planning" OR "regional governments" OR "regional healthcare" OR "area wide planning" OR "regional healthcare network" OR "comprehensive health planning" OR "annual implementation plans" OR "regional governments" OR "regionalization") AND ("decentralization" OR "coordination" OR "co ordination" OR "rationalization" OR "rationaliz$" OR "governance).

There was no language or date restriction, and the search was conducted between February and March 2017 and updated in October 2020.

The inclusion criteria were studies that answered the PICOT question. Studies concerning systematic reviews methodology, overviews of systematic reviews, theoretical, qualitative, editorial studies and letters to the editor were excluded, as well as studies that did not address interventions related to clearly-defined organizational arrangements of regionalization, or that did not provide the full text of the study.

The systematic review followed the PRISMA for systematic review protocols. The protocol was previously registered on the Prospero platform (CRD42016042314).

The selection of studies was performed in two steps: i) selection by title and abstract by four independent reviewers (MR, ENS, JOMB and HES), and the differences resolved by consensus; ii) selection after reading of the studies in full by two independent reviewers (MR and ENS), with disagreements resolved by consensus.

We also made changes to the results, to describe the 6 studies included with the new search strategy:

RESULTS

 The search found 3921 documents, of which 604 were duplicates. After reading the titles and abstracts, 486 documents were excluded. The reason for excluding this step is described in the Supplementary Material. After complete reading of the documents, 39 studies were selected to compose the systematic review. Figure 1 describes this process.

[…]

Table 1 summarizes the main characteristics of the studies included in this systematic review. Regarding the year of publication, there is a study published in the 1980s[42], six in the 1990s[30,31,33–35,38], seventeen in the 2000s[17–20,27–29,32,37,41,43,46,47,52,54–56], fourteen in the 2010s[22–24,26,36,39,40,44,45,48–51,53] and one in the 2020s[21]. As for the period studied, there is a wide variety. Studies[17,29,32,37,41,42,46,50,51] with up to one year of analysis corresponded to 23.07% (n=9) of publications, followed by those with more than 10 years[20,24,31,55] (10.25%; n=4) and those with one year of analysis[27,33,38,56] (10.25%; n=4). Regarding the country of origin, 56.41% (n = 22) are from the United States[18,20,22,23,27,29–31,35–39,41–43,45–47,52,55,56]. Publications from Italy[21,26,48] total 7.69% (n=3), while publications from Australia[32,51], Brazil[17,40], the Netherlands[50,54] and Canada[28,34] account for 5.12% (n=2) each. Zambia[19], Uganda[49] and a partnership between Canada and the United States[33], and between Italy and Spain[24], each of the publications with 2.56%. (n=1).

Regarding the study method, 11 distinct methods were identified, and the most frequent was the regression model[17,20–22,24,27,29–31,33,35–38,42,43,46,47,52–56], in general, with 58.97% (n=23). However, it is noteworthy that the regression models varied between simultaneous, Poisson, linear, generalized linear equations with Poisson error structure, logistical, logistical and linear, hierarchical, multiple and Cox multivariate logistics. Ecological[32,37,48] and cross-sectional studies[40,49,51] add up to 7.69% each (n=3) and quasi-experimental studies[44,45] accounted for 5.12% (n=2). The methods of cost-utility analysis[18], proportional risk survival analysis[28], cost-effectiveness analysis[23], cost analysis[50], Propensity score[41], multiple comparison test[34], cohort[39] and Data Envelopment Analysis[26] totaled 2.56% of the studies, each.

For quality assessment, the instrument of The Joanna Briggs Institute was used[57]. After weighting by two independent researchers, 58.97% of the studies obtained maximum quality assessment (8/8), while 23.07% obtained intermediate evaluation (7/8 or 6/8). Another 10.25% of the studies had regular evaluation (4/8) and 5.12% poor evaluation, with 0/8 points established by the instrument. The result of the quality assessment can be seen in detail in Table 2.

Also, minor changes were made, identified in the Marked-up copy. Four strategies were identified in the new studies included. Two studies addressed the rationalization dimension, with the strategy “Concentration of procedures in hospitals with high volume of production”. 

Regarding the coordination dimension, the “Coordinated care organization” strategy was included based on the results of the new studies:

Within the “coordination” dimension were the interventions of “Integration of surgery and cancer therapy in the same physical facility”, “Provision of sanitary transport services between areas further from the high-volume hospital”, “Coordinated care organization”, “Outreach services in remote areas” and “Outpatient and psychosocial network integration”.

[…]

The third intervention refers to coordinated care organization, which are patient-centered integrated health services that are geographically defined with a reference site for population care. They are services of low and medium complexity that act as a gateway to outpatient care. Two different studies have been identified for depicting this type of intervention[44,45]. 

The first one, carried out in France, performed a quasi-experimental study devised into two clusters: ‘everyday care’ and ‘coordinated care’ in the treatment of cancer patients who were on oral medication. The authors identified that the overall six-months survival rate was of 76% in the first cluster and of 87% in the latter (p = 0.064). No significant statistical changes were observed in the rates of disease progression, quality of life or treatment compliance[44].

The second study was conducted in the United States of America (USA). It analyzed the trends in early prenatal care initiation and trends in prenatal care adequacy after the implementation of the coordinated care organization model. The authors found that the rate of early prenatal care initiation increased significantly, from 73.1% to 77.3%, while prenatal care adequacy from 65.9% to 70.5%. The effect of implementing CCO in prenatal care adequacy, however, was not significant. After a sensitivity analysis, the estimates remained constant[45].

Regarding the decentralization dimension, a new study was included in the “Decentralization of care” strategy, being:

Six interventions were categorized as belonging to the “decentralization” dimension. The first, “Decentralization of care”, states that health care actions should be performed at the regional level, including four studies in this intervention.

[…]

The last study included in this intervention compared the health outcomes of patients who were tended by either traditional or decentralized teams. The analyzed supply variables showed that: patients who had been tended by decentralized teams experienced only half of the expected amount of treatment and of the annual outpatient doctor-patient interactions; but the amount of hospitalizations was 17.3% higher among patients who had been given decentralized care. The annual costs of hospitalizations ranged from 886 to 3223 Euros for decentralized teams and 2038 Euros for central outpatient care. The costs of psychotropic and outpatient treatment, however, are significantly higher for the central outpatient department (1,050 Euros per year for the former and 781 Euros for the latter). The authors conclude that the number of persons admitted has slightly increased, as did the number of admissions. Despite that, the number of admissions per person admitted to the hospital did not suffer any alterations[53].

Still in this dimension, the new strategy “Fiscal decentralization” was identified, including the excerpt:

The fifth intervention was called "fiscal decentralization" and refers to the fiscal autonomy of regional health authorities (RHA). It is a financing mechanism, a quasi-market model with purchaser-provider. This model allows RHA to be responsible for delivering a basic package of health care services through a network of population-based local health authorities as well as public and private accredited hospital providers. A study conducted a convergence analysis to assess whether fiscal decentralization could improve two selected health indicators: the infant mortality rate (IMR) and the increase in life expectancy at birth (LEB). The authors claim that there is no clear indication of σ convergence or long-term σ divergence for both outcomes in patients’ health. According to the authors, the effects of fiscal decentralization do not seem to demonstrate a systematic dynamic. 

For both outcomes in patients’ health, all axis exhibits a negative slope (always statistically significant at 1% level), which is consistent with the hypothesis of β-convergence. Furthermore, as the level of decentralization increases, the slope of the axis (β coefficient) increases. This indicates the beneficial effect of higher degrees of fiscal decentralization in improving the convergence process. However, the fact that the axis intersect in the analysis indicates that the decentralization growth ratio is sensitive (moderate) to the level of health outcomes of the geographical region being analyzed. As the IMR decreases and LEB increases, the fiscal decentralization’s efficiency in contributing to the reducing of the IMR or the increase of LEB is reduced[21].

In the DISCUSSION, two new paragraphs were included, which are transcribed below:

Despite the positive results of regionalization, Cavalieri & Ferrante (2020) warn that they have not identified any obvious connections between fiscal decentralization and improvements in a population’s health. This result is in agreement with the published literature on the subject. Di Novi et al. (2019) observed that, although fiscal decentralization helps to contain disparities between regions, there is no statistical difference in the analyzed models. The results indicate that wealthier health regions tend to obtain better results with fiscal decentralization, indicating the reduction of inequities in health. Underprivileged regions, however, continue to rely on subsidy at central level, which does not render any effect for the RHA in terms of accountability and governance[77]. 

Moreover, the study indicates that the significance of the impact of fiscal decentralization increases over the years[77], supporting the idea that it takes time to observe the effects of regionalization on a population’s health. This finding corroborates the need for further studies on health regionalization, since the topic still has critical gaps that have not yet been exhausted by its scientific literature.

Reviewer 1, comment 2: “Accordingly, they should also acknowledge adherence to the research guidelines for systematic reviews, PRISMA, as in http://www.prisma-statement.org/.

Our response: 

A new revision of the manuscript was carried out in order to comply with PRISMA for systematic review protocols.

Reviewer 1, comment 3: “One small comment regards the research question, for which I would recommend a slight rewording. "A systematic review was conducted to answer the following question: what are effective interventions to improve health and management indicators within the health services regionalization?" I recommend striking the word THE and instead it should read: "what are effective interventions to improve health and management indicators within health services regionalization?" I look forward to seeing the updates on this study after the revision of the search strategy.”

Our response: 

We changed the text, as suggested.

---

## [Editor Report · Decision Letter 2]

3 Dec 2020

Regionalization for health improvement: a systematic review

PONE-D-19-27817R2

Dear Dr. Ramos,

We’re pleased to inform you that your manuscript has been judged scientifically suitable for publication and will be formally accepted for publication once it meets all outstanding technical requirements.

Kind regards,

Itamar Ashkenazi

Academic Editor

PLOS ONE